# Pistol Shooting Performance Correlates with Respiratory Muscle Strength and Pulmonary Function in Police Cadets

Emre Karaduman [1], Özgür Bostancı [1], Fatih Karakaş [2], Menderes Kabadayı [1], Ali Kerim Yılmaz [1], Zeki Akyildiz [3], Georgian Badicu [4,*], Stefania Cataldi [5,*] and Francesco Fischetti [5]

1 Faculty of Sport Sciences, University of Ondokuz Mayıs, 55270 Samsun, Türkiye; karaduman.emre@hotmail.com (E.K.); bostanci@omu.edu.tr (Ö.B.); menderes@omu.edu.tr (M.K.); akerim.yilmaz@omu.edu.tr (A.K.Y.)
2 Faculty of Sport Sciences, University of Sinop, 57010 Sinop, Türkiye; fkarakas@sinop.edu.tr
3 Sports Science Department, Gazi University, 06500 Ankara, Türkiye; zekiakyldz@hotmail.com
4 Department of Physical Education and Special Motricity, Faculty of Physical Education and Mountain Sports, Transilvania University of Braşov, 500068 Braşov, Romania
5 Department of Basic Medical Sciences, Neuroscience and Sense Organs, University of Study of Bari, 70124 Bari, Italy; francesco.fischetti@uniba.it
* Correspondence: georgian.badicu@unitbv.ro (G.B.); stefania.cataldi@uniba.it (S.C.)

**Abstract:** Breathing patterns play a crucial role in shooting performance; however, little is known about the respiratory muscle strength and pulmonary capacities that control these patterns. The present study aimed to examine the relationship between shooting performance, respiratory muscle strength, and pulmonary function and to determine differences in respiratory capacities according to the shooting performance categories in police cadets. One hundred sixty-seven police cadets were recruited to assess respiratory muscle strength, pulmonary function, and shooting performance in a well-controlled environment. Measurements included maximal inspiratory pressure (MIP), maximal expiratory pressure (MEP), forced expiratory volume in 1 s ($FEV_1$), forced vital capacity (FVC), slow vital capacity (SVC), maximal voluntary ventilation (MVV), and pistol shooting scores. The shooting score had a moderate positive correlation with MIP ($\rho = 0.33$) and MEP ($\rho = 0.45$). FVC ($\rho = 0.25$), $FEV_1$ ($\rho = 0.26$), SVC ($\rho = 0.26$) ($p < 0.001$) and MVV ($\rho = 0.21$) ($p < 0.05$) were slightly correlated with shooting score. There were differences between shooting performance categories in MIP, MEP, FVC, $FEV_1$, SVC, and MVV ($p < 0.001$, $p < 0.05$). The results imply that both strong respiratory muscles and optimal pulmonary function may be one of the necessary prerequisites for superior shooting performance in police.

**Keywords:** inspiratory muscle strength; expiratory muscle strength; pulmonary capacity; police officers; shot performance

## 1. Introduction

Police officers responsible for ensuring the peace and security of societies are the sole agents of authorized violence in modern states [1]. These social missions can be complex, stressful, and tense as they are physically, emotionally, and socially challenging, as well as dangerous [1–5]. Numerous empirical studies report that people working in high-risk occupations such as police, military, and law enforcement face elevated stress levels, especially during an acute threat [6–8]. The officer's response to a critical incident (e.g., armed aggression) is often a life-threatening scenario for the police, suspect, and the public and requires effective and accurate intervention to ensure the safety of all involved [6]. For example, police may be asked to make a life-or-death decision based on their assessment of a particular situation or intervene in cases such as a high-speed chase or conflict with a suspect [6,7,9]. Undoubtedly, although lethal force is rarely used by police

officers when firearms are required [1,10], officers need to perform extremely well with respect to shooting accuracy.

Pistol shooting is a precision activity that requires physical skill as well as technical infrastructure and mental focus [11,12]. Previous studies revealed that significant performance deterioration occurs during stressful shooting moments involving psychological or physical stress [2,3,13–18]. Nieuwenhuys and colleagues [2] indicated that stress-induced deterioration in shooting performance resulted from decreased accuracy, faster reaction times, increased blink frequency (i.e., an increase in the amount of time the eyes are closed), and more false-positive decisions (i.e., shooting unarmed targets) [2,3,14,15]. Further, several authors have revealed that shooting performance is affected by many factors such as acute exercise-induced fatigue [18], increased heart rate [17], and breathing patterns [19]. An unsuccessful shooting performance during critical tasks (e.g., organized crime, terrorism, armed aggression, armed insurrection, civil war, etc.) may have tremendous consequences for both lives of the police themselves and everyone involved [6,7,20]. Thus, it is essential for competent law enforcement officers (e.g., military, police) to better understand and control the responsible factors that affect shooting performance in stressful encounters [6,21].

Several studies have suggested that acute psychological and/or physiological stressors in critical moments can rapidly reduce cognitive functions, as well as physical health and psychomotor performance, which ultimately result in altered shooting performance (accuracy, shooting time, postural control, stability of hold, etc.) [8,15,17,22–28]. When considering performance generally, psychological stress arousal can result in increased task errors and degradation of task accuracy [2,3]. Behavioral results in police studies show the detrimental effects of acute psychological stress on motor control or cognitive functions such as attention, perception, and decision-making [14,29–32]. Other detrimental effects are the serious stability threats caused by acute and intense psychological stress. Stress may modulate general motor patterns such as movements [33] and balance [34,35] by altering the emotional state [36–38]. Guillot and colleagues [39] suggested that control of emotional load is also necessary for high performance, especially immediately before shooting [39,40]. A superior shooting performance represents a motor performance skill that requires good postural stability [41–43] and fine motor control [44] under acutely stressful working conditions [24]. Therefore, assuming that executive functions are the first cognitive functions to suffer in stressful situations [45], it can be considered that acute stress in critical situations should be kept under control for shooting performance.

Physiological stress may impair performance as a result of a short-term acute dynamic action (e.g., chasing a fleeing suspect on foot) [26]. It has been reported that high cardiac functions (blood pressure or heart rate) significantly affect shooting performance in different facets [17], and deceleration of heart rate before shooting has been linked to optimal shooting performance [46]. Previously, studies have indicated that in some situations such as potential threats and activities that require the use of force, the cardiovascular stress response of police officers is increased [5,9]. The stress response has certain physiological effects on our neurobiological functions, causing, e.g., increased heart rate, vasodilation (widening of blood vessels), blood pressure, and breathing [47,48]. Physiologically, the first response of the autonomic nervous system to stress is to regulate the heart rate by stimulating the cardio-respiratory system [47]. Brisinda and colleagues [49] noted that short-term heart rate variability associated with operational stress might be affected by changes in respiratory patterns [49]. Although an increased heart rate is assumed to adversely affect performance [17,50], the effect of breathing patterns (i.e., deep or rapid breathing) on postural regulation may be the main factor responsible for deteriorated performance [19]. Lakie [51] reported that physical stress-induced increases in heart rate and ventilation further contribute to the tremor [51]. Likewise, it has been reported that acute physiological stress also affects stability functions such as balance or body sway through adjustment of cardiorespiratory function [52]. Ball, Best and Wrigley [53,54] found that the body sway factor is one of the most important successful shooting factors [53,54]. It has been

documented that a prerequisite for successful performance is good body stability, which contributes to minimal movement of the pistol barrel during the aiming phase [53–55].

Together, these factors demonstrate that stress can manipulate shooting performance and cause significant impairments to accuracy. Several studies have focused on understanding the effectiveness and utility of practices that help develop various coping skills and strategies such as reality-based exercises [13,56], stress training programs [57], or breath control [58,59], which can control—or even improve—stress-induced deterioration in police officers. Breathing control has been adopted in the military and martial arts to reduce autonomic stress responses and to maintain performance in the face of a threat [60]. The training guides known to the military and police officer trainer communities state that body relaxation and some breathing patterns are key components of firearm training [58,61,62].

Physiological functions such as liquid movements and cardiac and respiratory effects (muscle contractions) cause body sway [63–65]. Specifically, it is known that breathing movement interferes with the biomechanical properties of the trunk, which leads to minor perturbations within the body (e.g., breathing cycle) that may trigger a certain amount of sway [66–68]). Frazer [69,70] reported that uninterrupted breathing moves the diaphragm and chest muscles, which results in an unsteadiness of the body. Therefore, shooters have adopted fixation strategies such as a breath-hold while aiming and squeezing the trigger [69,70]. In a situation where shooters may need to fire quickly, under mental or physical stress, the heart rate and breathing increase as the lungs can demand more air. Under these circumstances, breath control involves simply stopping breathing and holding it. A stable shot is considered to be taken at the most optimal interval between inhalation and exhalation (in a natural pause in the breathing cycle), known as breath-hold. Good timing with respect to the breathing cycle can play a critical role in the stability and accuracy of a shot [71]. Hence, shooters release the shot by holding their breath in a period of a natural 2.5-s pause that occurs at the bottom of the exhalation [72]. Pausing breathing for a moment while the shot is being made stabilizes the position and allows a quick shot or an accurate series of shots [61]. Assuming that the diaphragm and abdominal muscles are one of the main factors in pausing breathing, maintaining the accuracy of the shot may depend on respiratory muscle strength [73,74]. However, the information regarding respiratory muscle strength in police officers is limited, and the relationship between shooting performance and respiratory capacities is still unclear. The present study examined the correlation between shooting performance and shooters' respiratory capacities, i.e., respiratory muscle strength and pulmonary function.

## 2. Materials and Methods

The participants visited the laboratory twice and shooting range once within a week, separated by one day. The first visit consisted of anthropometric assessments and familiarization with testing procedures. During the second visit, respiratory muscle strength and pulmonary function were measured. Shooting scores were evaluated at the last visit.

### 2.1. Participants

This study included 107 male and 60 female police cadets who volunteered from the 19 Mayıs Police Academy (Table 1). None of them were smokers and had no recent infection or respiratory disease history. Participants abstained from alcohol and caffeine 24 h prior to testing and arrived at the laboratory 2 h post-prandial. The local ethics committee approved the study (Ruling no: E-95674917-108.99-29362), and it was conducted according to the Declaration of Helsinki. All participants were fully briefed about the study and provided written informed consent.

**Table 1.** Participants' characteristics and shooting score (*n* = 167).

| | Low Score (45) | | Moderate Score (77) | | High Score (45) | |
|---|---|---|---|---|---|---|
| | **Male (24)** | **Female (21)** | **Male (48)** | **Female (29)** | **Male (35)** | **Female (10)** |
| Age, yr | 20 (19–21) | 19 (19–20) | 20 (19–21) | 19 (19–21) | 20 (19–21) | 20 (19–21) |
| Weight, kg | 72 (67–77) | 61 (55–67) | 70 (66–76) | 65 (60–70) | 71 (65–76) | 63 (58–68) |
| Height, cm | 175 (170–180) | 163 (162–165) | 174 (170–180) | 165 (164–168) | 176 (171–179) | 166 (164–171) |
| BMI, kg/m$^2$ | 24 (22–25) | 23 (20–25) | 23 (22–25) | 24 (21–26) | 23 (22–25) | 23 (21–24) |
| Shooting score * | 130 (122–140) | 121 (106–131) | 166 (157–171) | 160 (148–169) | 184 (179–189) | 184 (177–187) |

* $p < 0.05$ between categories within same-gender. Data are presented as median and interquartile ranges (25th and 75th percentiles). BMI: Body mass index.

### 2.2. Anthropometric Measurements

A calibrated electronic scale was used to assess body mass (with sports clothes, without shoes) and height (anatomical position) (SECA, Hamburg, Germany). Measurements were recorded to precision to 0.1 cm and 0.01 kg, respectively. Body mass index (BMI) was calculated by dividing body weight in kilograms by height in meters square (BMI = kg/m$^2$).

### 2.3. Respiratory Muscle Strength

Maximum inspiratory pressure (MIP) and maximal expiratory pressure (MEP) tests were performed to assess the respiratory muscles strength. The measurements were carried out using a MicroRPM (CareFusion Micro Medical, Kent, UK) and assessed in accordance with available guidelines in the field [75]. MIP was measured after full expiration; participants were instructed to exhale to residual volume and then inhale as hard and as fast as possible to total lung capacity. MEP was performed after full inspiration; participants were instructed to inhale to total lung capacity and then exhale with maximum force. The measurements were performed in an upright sitting position using a nose clip. At least three trials were performed before the highest value was recorded. A one-minute rest period was allowed between each trial [76].

### 2.4. Pulmonary Function Test

Pulmonary functions were determined using forced expiratory volume in 1 s (FEV$_1$), forced vital capacity (FVC), slow vital capacity (SVC), and maximal voluntary ventilation (MVV) by spirometry (CPFS/D USB Spirometre, MGC Diagnostics, Saint Paul, MN, USA, ABD) assessed in accordance with available guidelines in the field [77]. Participants completed a minimum of three satisfactory maneuvers with the highest recorded. A one-minute rest period was allowed between each trial [76].

### 2.5. Shooting Task

Police cadets used a validated 9.0 mm caliber pistol (SAR9) (Sarsılmaz Pistol Industry Co. Inc., Düzce, Türkiye) and equipment such as ear and eye protection provided by the relevant unit administration. Participants made 20 shots from a distance of 10 m, following the safety rules under the supervision of shooting instructors. Before starting the shooting series, sighting and warm-up shots were made. All participants shot with one hand, without support, and standing. There was no time limit (20 shots, 20 ± 1 min). To standardize the tests, the maximum feet distance was kept between 30 (narrowest) and 60 cm (widest), and Hawkins and Sefton's guidelines were followed [78].

A standard, light-colored cardboard target with concentric score zones (innermost zone worth 10 points) was used, and the shot score was reported after each shot. The effectiveness of the pistol shot was evaluated by the total scores (maximum score, 200) of the bullets hitting the cardboard target. Shooting conditions were set up on an indoor shooting range in accordance with the official rules and regulations of the International Shooting Sports Federation [79].

*2.6. Statistical Analysis*

Statistical Package for the Social Sciences (SPSS) 22.0 (IBM Inc., Chicago, IL, USA) was used for statistical analyses, and a two-sided $p$-value < 0.05 was considered as statistical significance. Descriptive data were presented as median and interquartile range (IQR). We divided participants into quartiles based on shooting scores [80], categorizing Q1 as a low-score (LS) ($\leq$143 points), Q2–Q3 as a moderate score (MS) (144–176 points), and Q4 as a high score (HS) ($\geq$177 points). Data distributions were checked for normality using a Kolmogorov–Smirnov test. The Kruskal–Wallis H tests was used for comparisons. Post-hoc comparisons were made using Tamhane tests. Spearman's rho ($\rho$) correlation was employed to assess the association between shooting scores and respiratory capacities ($p < 0.05$). A correlation ($\rho$) of $\geq$0.10 was considered small, $\geq$0.30 moderate, and $\geq$0.50 large [81]. All graphics were made using GraphPad Prism software (version 9.2.0, GraphPad Software Inc., San Diego, CA, USA).

## 3. Results

Participant characteristics are presented in Table 1. There were no overall differences in age, weight, height, and BMI among the HS, MS, and LS categories within the same gender ($p > 0.05$), while there was a difference in shooting scores ($p < 0.05$).

Table 2 shows the correlation between shooting scores and respiratory functions according to gender. Although shooting scores in both males and females showed a significant positive correlation with MIP and MEP ($p < 0.05$), there was no association with $FEV_1$, FVC, $FEV_1$/FVC, SVC, and MVV ($p > 0.05$).

**Table 2.** The correlation between shooting score and respiratory functions according to gender.

| | Male (107) | | Female (60) | |
|---|---|---|---|---|
| | **Shooting Score** | | | |
| | $\rho$ | *p*-Values | $\rho$ | *p*-Values |
| MIP (cmH$_2$O) | 0.455 | **<0.001** | 0.213 | **0.028** |
| MEP (cmH$_2$O) | 0.473 | **<0.001** | 0.321 | **0.001** |
| $FEV_1$ (L) | 0.092 | 0.484 | 0.062 | 0.527 |
| FVC (L) | 0.027 | 0.841 | 0.103 | 0.291 |
| $FEV_1$/FVC (%) | −0.168 | 0.199 | 0.055 | 0.577 |
| SVC (L) | 0.000 | 0.998 | 0.153 | 0.115 |
| MVV (L/min) | −0.202 | 0.122 | 0.108 | 0.268 |

$FEV_1$; Forced vital capacity at 1 s; FVC: Forced vital capacity; MEP: Maximal expiratory pressure; MIP: Maximal inspiratory pressure; MVV: Maximal voluntary ventilation; SVC: Slow vital capacity; $\rho$: rho.

The shooting score had a moderate positive correlation with MIP ($\rho = 0.33$) and MEP ($\rho = 0.45$) (Figure 1). However, FVC ($\rho = 0.25$), $FEV_1$ ($\rho = 0.26$), SVC ($\rho = 0.26$) ($p < 0.001$) and MVV ($\rho = 0.21$) ($p < 0.05$) had only a small correlation with shooting score (Figure 2). No significant correlation was observed with $FEV_1$/FVC ($p > 0.05$).

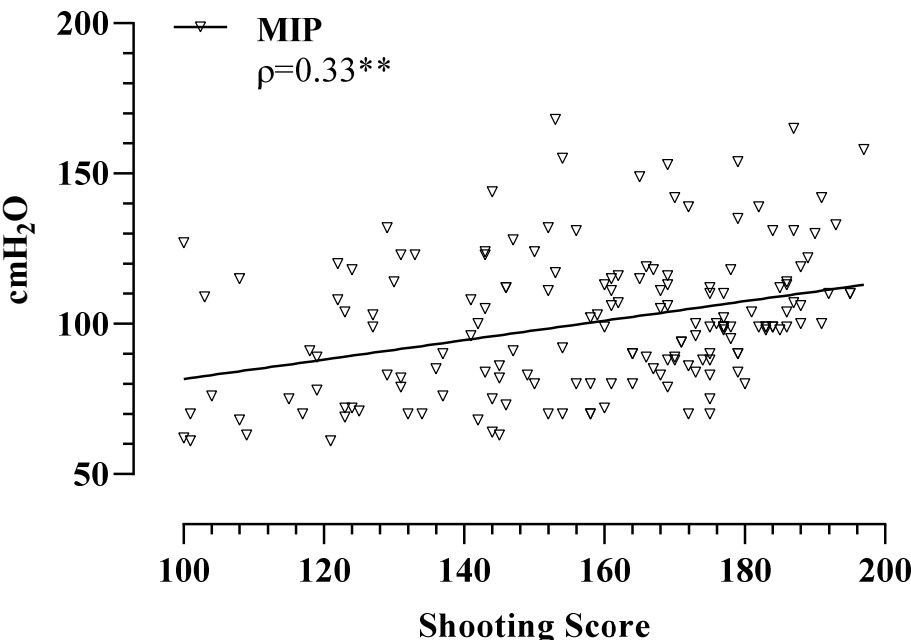

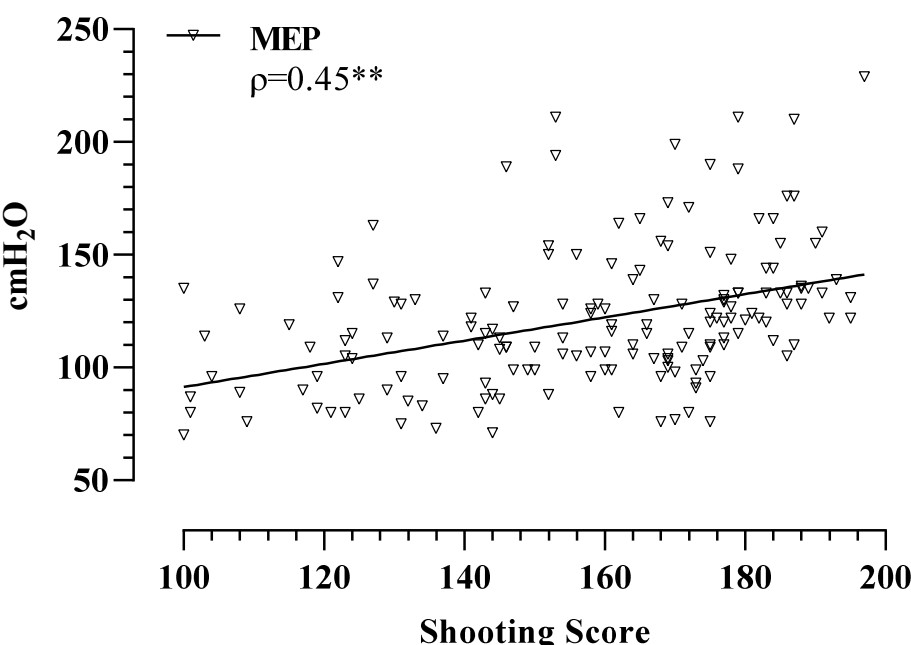

**Figure 1.** Scatterplots with regression lines and correlation coefficients between shooting score and maximal inspiratory pressure (MIP), and maximal expiratory pressure (MEP). ** $p < 0.001$. Data are presented as correlation coefficients ($\rho$). MEP: Maximal expiratory pressure; MIP: Maximal inspiratory pressure.

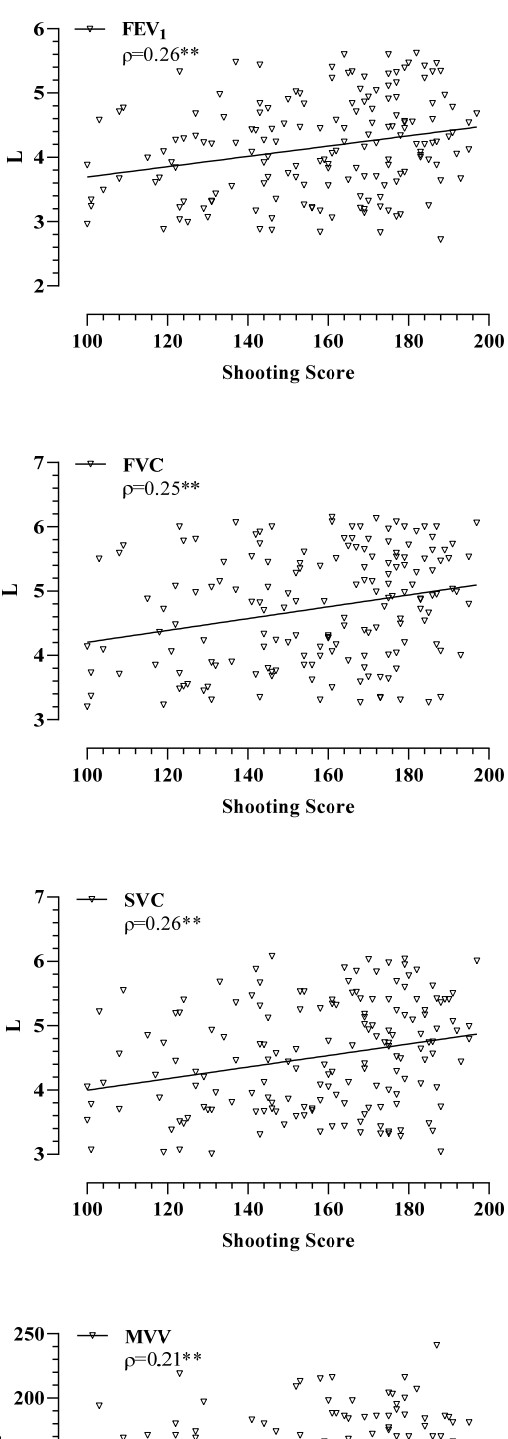

**Figure 2.** Scatterplots with regression lines and correlation coefficients between shooting score and pulmonary function parameters. ** $p < 0.001$. Data are presented as correlation coefficients ($\rho$). FEV$_1$; Forced vital capacity at 1 s; FVC: Forced vital capacity; MVV: Maximal voluntary ventilation; SVC: Slow vital capacity.

## 4. Discussion

In the present study, we examined the relationship between shooting performance and respiratory capacities in police cadets. The current study results demonstrated that all respiratory muscle strength and pulmonary function parameters correlated with shooting performance (Figures 1 and 2). In addition, we reported that the high-scoring police cadets had the highest respiratory muscle strength and pulmonary function parameters (Figure 3).

Following a physiological stressor, metabolic activation increases the cardiac and breathing rhythm [47,82], which cause body sway [19]. Pendergrass et al. [83] showed increased postural sway after a 2-mile run completed at 93 percent of maximum heart rate [83]. Although the body may respond differently depending on the intensity, nature, and duration of the stressor, as well as several internal factors of the individual who experience it [84,85], body sway is a highly sensitive action and is affected by stability muscles. The respiratory muscles act as one of the spinal stability muscles [86] and optimize postural sway [68,73,87]. Studies have reported that the diaphragm and abdominal muscles produce a hydraulic effect that aids spinal stabilization by stiffening the lumbar spine with increased intra-abdominal pressure in the abdominal cavity [88]. Undoubtedly, respiratory muscles not only contribute to postural sway but also to optimal fixation strategies such as breath control (breath-holding, diaphragm breathing, etc.). Breathing techniques such as deep breathing, diaphragmatic breathing, or tactical breathing, which assist in maintaining breathing control, are rather popular with law enforcement personnel [25]. The purpose of breathing control is to provide more stable alignment and stabilization when aiming and squeezing the trigger. In the best timing and duration, breath-holding according to the breathing cycle can contribute to the shot's accuracy [71]. Further, breath-holding provides more hold stability as it stabilizes the chest, abdomen, and shoulder movement. Therefore, the respiratory muscles are important in providing stability to minimize unwanted movements during shooting, albeit with a different mechanism [42]. Some reports show that an increase in the strength and endurance of these muscles might increase spinal stability [86,89].

Several studies in patients with respiratory or neurologic diseases have shown that weakness in respiratory muscles may reduce body stability [90,91]. Likewise, recent studies have noted reduced balance and coordination in individuals with lung dysfunction [92–94]. Kocjan et al. [68] proposed that diaphragm weakness creates a state of insufficient irritability of proprioceptors and subsequent inadequate sensory stimuli to maintain optimal postural control and stability [68]. The most promising explanation for the relationship between respiratory muscle strength and shooting performance may be that the respiratory muscles responsible for stability contribute to holding ability, accuracy, and precision by providing a more stabilized posture during shooting.

As for pulmonary function, $FEV_1$, FVC, SVC, and MVV had a small correlation with shooting performance. To our knowledge, no study has examined the relationship between pulmonary capacities and pistol shooting performance, but indirect effects have been demonstrated in different breathing patterns or ventilation levels. Breathing patterns and postural control are related because even with quiet breathing, the ribs' rapid/slow and deep movement may cause postural instability [95]. Hamaoui et al. [67,96] showed that a thoracic breathing mode induces a disturbing effect on posture [67,96]. On the other hand, Malakhov et al. [97] indicated that different ventilation levels could impair standing posture in different ways and, as a consequence, could induce different postural control responses [97]. As these breathing responses depend on the minute volume and increase with the respiratory flow, perhaps optimal respiratory functions are able to compensate for increases in body sway.

Moreover, pulmonary capacities may affect the time of the breath-holding phase during shots. In scenarios with high-intensity dynamic actions, it may be difficult to hold one's breath [25]. In the marksmanship training guidelines [72], it is stated that the most optimal interval in the respiratory cycle for shooting is the natural pause of 2.5 s at the bottom of expiration, but sometimes this time may be insufficient to confirm the sight

picture and release the shot. Shooters may increase the natural pause in the respiratory cycle to approximately 8.5 s with several deep inspiration and expiration patterns [72]. However, breathing should not be stopped for too long in those with insufficient pulmonary capacity because the lack of oxygen can cause blurred vision and trembling muscles, which may adversely affect the performance of cognitive skills and visual acuity [59,71]. Therefore, another possible explanation for the relationship between pulmonary functions and shooting performance may be due to the fact that pulmonary capacity can affect breath-holding time. Some previous studies demonstrate links between shooting performance and lung capacities in archers [98,99]. However, another study showed no correlation between air pistol shooting performance and pulmonary function in shooters competing in the young female category (age 16) [12].

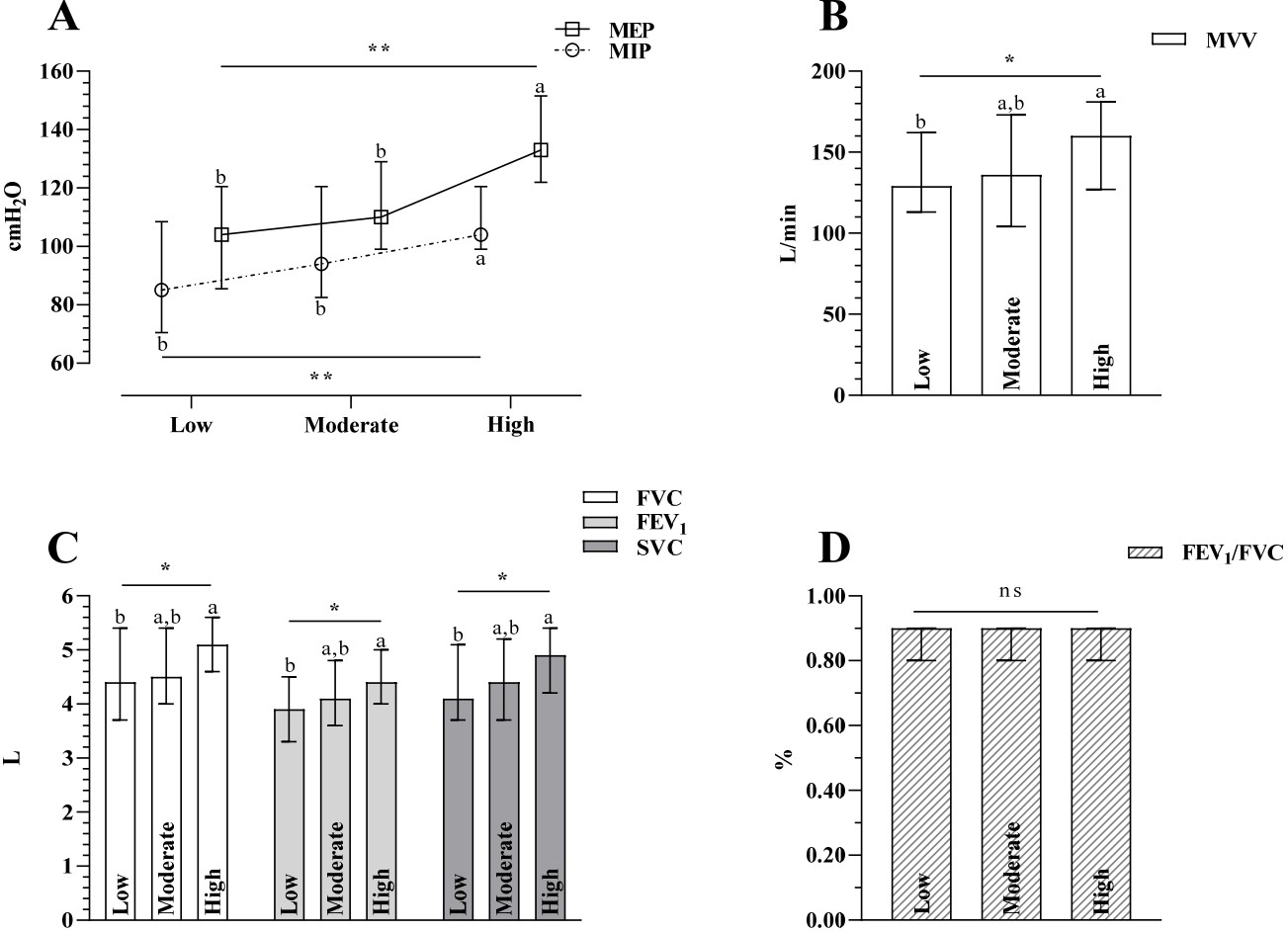

**Figure 3.** Comparison of respiratory muscle strength (**A**) and pulmonary function (**B–D**) according to shooting performance categories (Low score$_{(n)}$ = 45, Moderate score$_{(n)}$ = 77, High score$_{(n)}$ = 45). $^{a,b}$ No significant differences between groups that share the same letter. ** $p < 0.001$, * $p < 0.05$ between categories. Data are presented as median and interquartile ranges (25th and 75th percentiles). FEV$_1$; Forced vital capacity at 1 s; FVC: Forced vital capacity; MEP: Maximal expiratory pressure; MIP: Maximal inspiratory pressure; MVV: Maximal voluntary ventilation; ns: non-significant; SVC: Slow vital capacity.

Furthermore, police cadets with high-scoring shooting performance had the greatest respiratory muscle strength and pulmonary function. The strong respiratory muscles may be maintaining more stable shoulder, spine, and pelvic girdle fixations by optimizing the postural attitude of the thorax. In addition, high pulmonary capacity can contribute to a stable shot by regulating breath-holding time. Presumably, these stability functions allow the

shooter to perform more effectively by providing more stable precision (i.e., consistency), accuracy (i.e., deviation), and stability of hold (i.e., pistol movement) in pistol shots.

The limitation of the present study is the absence of balance or body sway measurements. We decided to focus our study on shooting performance and respiratory capacities because recent studies have already confirmed the relationship between shooting performance and postural responses [41,42]. However, we still cannot rule out the possibility of an existing relationship between shooting performance, respiratory capacities, and balance responses. Thus, it is important to underline that future studies are necessary to enlighten the relationship between these factors.

## 5. Conclusions

The present study showed a correlation between shooting performance, respiratory muscle strength, and pulmonary functions. Furthermore, police cadets with high shooting performance had the highest respiratory capacities. The results imply that both strong respiratory muscles and optimal pulmonary function may be one of the necessary prerequisites for superior shooting performance in police. We believe that these results can be useful to trainers in police academies and shooting sports, enabling them to understand the pulmonary capacities that are important in shooting performance and guide the planning of training programs concerning shooting performance. Additionally, further study is needed to understand the importance of respiratory capacity in selecting excellent police who will exhibit effective shooting performance in potential intervention situations.

**Author Contributions:** Conceptualization, E.K. and Ö.B.; methodology, Ö.B., F.K. and M.K.; software, A.K.Y.; validation, E.K., Ö.B. and Z.A.; formal analysis, E.K.; investigation, M.K.; resources, Ö.B., F.K. and A.K.Y.; data curation, A.K.Y. and Z.A.; writing—original draft preparation, Ö.B. and Z.A.; writing—review and editing, E.K., F.K., Z.A., G.B., S.C. and F.F.; visualization, E.K., G.B. and S.C.; supervision, F.F.; project administration, E.K. and Ö.B.; funding acquisition, Ö.B., F.K. and F.F. All authors have read and agreed to the published version of the manuscript.

**Funding:** This research received no external funding.

**Institutional Review Board Statement:** The study was conducted in accordance with the Declaration of Helsinki, and approved by the Local Institutional Review Board (Code: E-95674917-108.99-29362).

**Informed Consent Statement:** Informed consent was obtained from all subjects involved in the study.

**Data Availability Statement:** The datasets used and/or analyzed during the current study are available from the corresponding author upon reasonable request.

**Acknowledgments:** The authors thank all the athletes who participated in the study.

**Conflicts of Interest:** The authors declare no conflict of interest.

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
