# Peer review of "Pistol Shooting Performance Correlates with Respiratory Muscle Strength and Pulmonary Function in Police Cadets"

_sustainability, doi:10.3390/su14127515_

Round 1
Reviewer 1 Report
Line No.
25 single digits should be written - one
104 separate words - short term
111 - Since there are two articles authored by Ball for 2003, it is best to include all authors first before using et al. Generally is is acceptable to list all first anyway.
173 - Use an adjective before introducing a number to begin a sentence.
232 - Write full wording for acronyms that begin a sentence.
259 - write single digit - one
278 - write the word "percent" instead of using the symbol when presenting in text.
384 - remove the colon
In addition to using italics for journal titles, it might be good to put the volume number in italics to differentiate from an issue number.
Author Response
Dear Reviewer,
I attach the answers to the comments.
Thank you!

Reviewer 2 Report
Comments to Authors
The study aims to evaluate the relationship between pulmonary capacity/function and shooting performance
The introduction is well written and provides a rationale for the study. However, it is unnecessarily too long. For instance, lines(64-67) (87-97)and (116-129) discuss the psychological effects on breathing patterns that can affect shooting performance. Details in these sections deviate from the purpose of the study since the psychological impact on performance is not examined. On the other hand, statements in lines 146-152 indicate that strong lung muscles can provide stability to the upper body and improve performance are to the point and can enhance the rationale.
Methods
Similarly to the introduction, the methods section was well written. The section includes details on all methods used to collect the data and is informative enough to allow replication. My main concern here is the fact that the participants were inexperienced. Shooting performance is affected by experience and practice; thus, how can you isolate the lung function on performance? That said, wouldn’t be better if you used regression models to see how the variability in lung function accounts for the variability in performance?
Stats
Since the shooting scores are affected by gender (line242), why have you performed the correlations with the whole group? I think that since females will have a reduced lung capacity due to smaller body sizes, that would also affect the correlations. To avoid bias, I believe it is better to demonstrate the associations for each gender.
Results.
Figure legends 1 and 2 have the statement “. Data are presented as correlation coefficients (p). Shouldn’t the parenthesis have the symbol rho(ρ) in it?
Discussion
Similarly to the introduction and the methods, the discussion section was well written. There is a sustainable relationship between the results and the related literature – very good fitting of these results with other findings. The study's major finding is that high-scored police cadets had the highest respiratory muscle strength and pulmonary function parameters. The major concerns about this study are the ones that were highlighted in the previous sections. Perhaps if gender is separated, the associations would be weaker.
Author Response
Dear Editor,
I attach the answers to the comments.
Thank you!

Round 2
Reviewer 2 Report
The authors have responded to the comments sufficiently. The only issue I see is that the conclusion section is too absolute. The results of the study indicate some association between shooting performance and pulmonary function. The last sentence indicates that "In addition, the study
may suggest that respiratory capacity may be one of the physiological criteria for selecting excellent police cadets who will exhibit effective shooting performance in potential intervention situations". I suggest you replace that by saying that further investigation is needed to support the aforementioned statement.
Author Response
Dear Reviewer,
I attach the answer to the comment.
Thank you!
